# Inhibitory Effect of Polypeptides Produced by *Brevibacillus brevis* on Ochratoxigenic Fungi in the Process of Pile-Fermentation of Post-Fermented Tea

**DOI:** 10.3390/foods11203243

**Published:** 2022-10-17

**Authors:** Zhenjun Zhao, Lingling Zhang, Yougen Lou, Yan Luo, Xianchun Hu, Xueli Pan, Huawei Wu, Jianjie Li, Huiling Mei, Xinghui Li

**Affiliations:** 1College of Horticulture and Gardening, Yangtze University, Jingzhou 434025, China; 2School of Foreign Studies, Yangtze University, Jingzhou 434025, China; 3College of Life Science, Yangtze University, Jingzhou 434025, China; 4College of Resources and Environmental Sciences, Nanjing Agricultural University, Nanjing 210095, China; 5College of Horticulture, Nanjing Agricultural University, Nanjing 210095, China

**Keywords:** polypeptide, ochratoxigenic fungi, *Brevibacillus brevis*, anti-fungal, *A. carbonarius* H9

## Abstract

Contamination by ochratoxigenic fungi and its prevention during the pile-fermentation of post-fermented tea have always been a concern. The present study aimed to elucidate the anti-fungal effect and mechanism of polypeptides produced by *B. brevis* DTM05 (isolated from post-fermented tea) on ochratoxigenic fungi, and to to evaluate their use in the pile-fermentation process of post-fermented tea. The results showed that polypeptides (produced by *B. brevis* DTM05) with a strong antifungal effect against *A. carbonarius* H9 mainly had a molecular weight between 3 and 5 kDa. The Fourier-transform infrared spectra of this polypeptide extract showed that it was a mixture consisting mainly of polypeptides and small amounts of lipids and other carbohydrates. The polypeptide extracts significantly inhibited the growth of *A. carbonarius* H9, and its minimum inhibitory concentration (MIC) was 1.6 mg/L, which significantly reduced the survival rate of spores. The polypeptides also effectively controlled the occurrence and ochratoxin A (OTA) production of *A. carbonarius* H9 on the tea matrix. The lowest concentration of polypeptides that significantly inhibited the growth of *A. carbonarius* H9 on the tea matrix was 3.2 mg/L. The enhancement of the fluorescence staining signal in the mycelium and conidiospore showed that the polypeptides with a concentration of more than 1.6 mg/L increased the permeability of the mycelium membrane and conidial membrane of *A. carbonarius* H9. The significant increase in the extracellular conductivity of mycelia suggested the outward leakage of intracellular active substances, and also further indicated an increase in cell membrane permeability. Polypeptides with a concentration of 6.4 mg/L significantly down-regulated the expression level of the polyketide synthase gene related to OTA production (*acpks*) in *A. carbonarius* H9, which may be the fundamental reason why polypeptides affect OTA production. In conclusion, reasonable use of the polypeptides produced by *B. brevis* can destroy the structural integrity of the cell membrane, make the intracellular active substances leak outward, accelerate the death of fungal cells and down-regulate the expression level of the polyketide synthase gene in *A. carbonarius*; thus, they can effectively control the contamination of ochratoxigenic fungi and OTA production during the pile-fermentation of the post-fermented tea.

## 1. Introduction

Post-fermented tea, also known as dark tea, is a unique and popular tea with a special aged-aroma, and a mellow, sweet taste that is mainly produced in Yunnan, Hunan, Hubei, Sichuan and other regions in China. The process of making post-fermented tea generally includes fixation, rolling, post-fermentation and drying [1]. As a result of the different production areas and different processing technology, post-fermented tea products are mainly divided into categories including Yunnan Pu-erh tea, Hubei Qingzhuan brick tea, Hunan Fuzhuan brick tea, Sichuan Tibetan tea and so on [2].

Pile-fermentation is a critical process in post-fermented tea production, in which a large number of microorganisms depend on a suitable temperature, humidity and nutrient conditions to grow and reproduce while simultaneously making a beneficial contribution to the unique flavor of post-fermented tea through their activities [3]. However, during the process of pile-fermentation, naturally inoculated microorganisms, especially some unknown fungi, grow in the tea and release their harmful metabolites, which brings potential risks to the safe drinking of post-fermented tea [4]. Currently, there are at least 19 known genera of fungi that have been isolated and identified from the process of pile-fermentation such as *Aspergillus*, *Penicillium*, *Rhizomucor*, *Trichoderma*, *Cladosporium*, *Fusarium*, *Talaromyces*, etc. [1,5,6]. It has been proved that several species from those genera are capable of delivering mycotoxins. Many potential risks concerning fungus in the process of pile-fermentation have been reported and even mycotoxins such as Aflatoxins, ochratoxin A, and fumonisins in tea have attracted a lot of attention [7].

Among the mycotoxins mentioned above, ochratoxin A (OTA) is a very common and deleterious toxicant that is mainly produced by the common genera of Aspergillus, Penicillium [8,9]. A recent study by our research group reported that 100% of *A. carbonarius* strains, *P. verrucosum* strains and *A. ochraceus* strains, 8.33% of *A. niger* strains and 30% of *A. tubingensis* strains isolated from post-fermented tea have the ability to produce ochratoxin A (OTA). Among ochratoxigenic fungi, *A. carbonarius* was identified as one of the most powerful OTA-producers (OTA production reached 55.56 ± 4.64 ng/g in Czapek yeast extract agar (CYA), exceeding that of other ochratoxigenic fungi) [10] and it is often found in the pile-fermentation process of post-fermentation tea along with other ochratoxigenic fungi [11]. Ochratoxigenic fungi can grow normally and produce ochratoxin A at temperatures of 25~40 °C and relative humidity over 40% [12,13], which is consistent with the temperature and humidity conditions of pile-fermentation [14]. Therefore, it is necessary to consider the potential safety risks caused by the growth of ochratoxigenic fungi and the ochratoxin A released in post-fermented tea; in particular, more attention should be paid to how to inhibit the growth of these fungi during the pile-fermentation process.

In the last few decades, chemical preservatives and chemical fungicides have been widely used as an effective and common means of controlling saprophytic or harmful fungi in the process of food production and the harvest stage of agricultural products [15]. However, because tea is a natural and healthy drink and the quality of post-fermented tea is inseparable from the participation of microorganisms originating from the complex microecosystem in the process of pile-fermentation, it is impossible to use chemical fungicides to inhibit the growth of harmful fungi during pile-fermentation of the post-fermented tea. Biological control may be one of the only control measures against harmful fungi in the pile-fermentation process. The *Bacillus* sp. have potential anti-fungal ability (by producing antifungal enzymes, polypeptides or other metabolites) [16,17,18] and have been found to be widespread in post-fermented tea. Many *Bacillus* species such as *B. subtilis*, *B. cereus*, *B. amyloliquefaciens*, *B. licheniformis*, *B. pumilus*, *B. brevis*, *B. coagulans* and *B. laterosporus*, etc., have been isolated and identified in post-fermented tea [10,19]. *B. subtilis* AF1 can inhibit the growth of *A. niger* by destroying the cell wall of *A. niger* [20]. *B. licheniformis* BL350–2 isolated from raspberry and strawberry jams inhibited the growth of mycotoxigenic species such as *A. carbonarius*, *A. ochraceus*, *Penicillium verrucosum*, etc. [21]. In our previous study, six different species of *Bacillus (B. substilis* DTM01, *B. coagulans* DTM02, *B. pumilus* DTM04, *B. brevis* DTM05, *B. licheniformis* DTM06 and *B. laterosporus* DTM03) were able to inhibit the growth of ochratoxigenic fungi on the tea matrix during the first 7 days of pile-fermentation. Among those, *B. brevis* DTM05 was found to be the strain with the highest anti-fungal activity because it can produce antimicrobial polypeptides with a molecular weight ≤ 10 kDa [10], which is similar to the characteristics (molecular weight and anti-fungal properties) of peptide produced by *B. brevis* GM100 (named Bac-GM100 with a molecular weight of 4375.66 Da) reported by Ghadbane et al. (2013) [22]. Although pure antimicrobial peptides have better antibacterial effects, they are limited in practical application due to their lower yields and higher costs. Therefore, in practice, the peptide mixtures may have more practical application.

The purpose of the present study was to investigate the anti-fungal effect and mechanism of a *B. brevis* peptides mixture against the ochratoxigenic fungus *A. carbonarius* during the pile-fermentation process of post-fermented tea in order to contribute to improving the process of pile-fermentation and reduce the safety risks associated with ochratoxigenic fungi in post-fermented tea.

## 2. Materials and Methods

### 2.1. Strains, Materials and Reagents

*B. brevis* DTM05 (Accession: MG986217) was isolated from post-fermented tea samples, identified by the amplification sequence of 16S rRNA in a previous study [10] and preserved by our laboratory. *A. carbonarius* H9, also from a previous study [10], was used as the representative strain of OTA-producing fungus isolated from tea samples in the pile-fermentation process of post-fermented tea for subsequent anti-fungal experiments. The preparation of *A. carbonarius* H9 spore suspension was as follows: *A. carbonarius* H9 were inoculated in potato dextrose agar medium (PDA) (potato, 200 g; glucose, 20 g; agar, 18 g and distilled water, 1 L) and cultured at 25 °C for 7 days. *A. carbonarius* H9 spores were washed with sterile water to prepare a high concentration of spore suspension. Then, the suspension was adjusted to the desired concentration with sterile water. The fungal spores were counted by the blood-cell counting-chamber [23].

The raw tea material was post-fermented tea provided by Hubei Xianning Dongzhuang Tea Industry Co., Ltd. SYTOX Green (SG) fluorescent stain and propidium iodide (PI) stain were obtained from the Invitrogen Company, CA, USA. Fungal fluorescent stain (main ingredient: calcofluor white) was purchased from Guokang Industrial Co., Ltd. (Zhaozhuang, China).

### 2.2. Instruments and Equipment

The following equipment was used in the study: a vertical automatic pressure steam sterilizer, Chongqing Simite Technology Co., LTD; SW-CJ-1F vertical super clean bench, Sujing Group Antai Co. LTD, Suzhou, China; Dhp-9082 Electric Thermostatic Incubator Shanghai Qixin Scientific Instrument Co., LTD, Shanghai, China; SYNERGYH1MG Auto ELISA detector, BioTek, USA; XSP-63B Fluorescence Microscope, Shanghai Optical Instrument No. 1 Factory, Shanghai, China; Avanti TM J-30I high-speed refrigerated centrifuge, Beckman Coulter, Inc. USA; DDS-307A conductivity meter, Shanghai Yi Electrical Scientific Instrument Co., LTD, Shanghai, China and a Nicolet iS10FT-IR, Thermo Fisher Scientific Inc., Madison, WI, USA.

### 2.3. Preparation and FT-IR Spectral Evaluation of Polypeptides (Molecular Weight from 3 kDa to 5 kDa) of B. brevis

Anti-fungal polypeptide (with a molecular weight ranging from 3 to 5 kDa) from the *B. brevis* strain was prepared as described by Che et al. (2015) [24] and Zhao et al. (2021) [10] with slight modifications (as follows). The *B. brevis* strain was cultured in LB medium (lysogeny broth: tryptone, 10 g; yeast extract, 5 g; NaCl, 10 g and distilled water, 1 L) at 37 °C in a shaker at 200 rpm for 5 days. The supernatant, taken from the culture medium after centrifugation at 8000 rpm, 4 °C, for 20 min, was filtrated with 0.22 μm filter membranes (Millipore, the Life Sciences branch of the Merck Group, Darmstadt, Germany) and the cells of *B. brevis* were discarded. The culture supernatant was precipitated with 60~70% (*w*/*v*) ammonium sulfate solution and the precipitates were recovered by centrifuging at 8000 rpm for 20 min at 4 °C. Thereafter, the precipitate was dissolved in a 20 mM phosphate buffer of pH 7.4 and then ultra-filtered with ultrafiltration centrifugal tubes of 3 and 5 kDa (Millipore, Darmstadt, Germany). The anti-fungal polypeptide solution with a molecular weight between 3~5 kDa was taken and freeze-dried at −20 °C as a reserve.

The anti-fungal polypeptide solution was diluted to different concentrations according to the polypeptide content, which was determined by Bradford G-250 reagent, for subsequent anti-fungal experiments.

The polypeptide freeze-dried powder was used for Fourier-transform infrared (FT-IR) spectroscopy measurement. For the specific method, we referred to Wang et al. (2021) [25].

### 2.4. Inhibition Effects of Polypeptides on the Growth of A. carbonarius H9

The inhibitory effects of the anti-fungal peptide mixture (with a molecular weight from 3 to 5 kDa) on the growth of *A. carbonarius* H9 were assessed according to Wang et al. (2018) [26] with some modifications. First, 180 μL 1 × 10^4^ CFU/mL spore suspension of *A. carbonarius* H9 (containing potato dextrose broth (PDB) of 5% total volume) was mixed with 20 μL anti-fungal peptide and loaded into the 96-well microplates. The final concentrations of anti-fungal peptide in the 96-well microplates were 0.1, 0.2, 0.4, 0.8, 1.6, 3.2, 6.4, 12.8, 25.6 and 51.2 mg/L. The control group was treated with sterile water instead of anti-fungal peptide. The mixture was cultured in an incubator at 25 °C for 48 h. The OD_600_ of the mixture was determined by using a microplate reader. The lowest inhibitory concentration that could still completely inhibit the growth of *A. carbonarius* H9 for 48 h was taken as the minimum inhibitory concentration (MIC).

### 2.5. Inhibition of A. carbonarius H9 on Sterilized Tea by Polypeptides

Fifty grams of the tea samples (water content 35%), previously sterilized at 120 °C for 15 min, were put into 250 mL tissue culture bottles, then 5 mL of a mixture containing polypeptides and spores of *A. carbonarius* H9 at a concentration of 1 × 10^4^ CFU/mL were added to the tea leaves in the treatment group. Different concentrations of antimicrobial peptides, 1.6, 3.2, 6.4, 12.8, 25.6 mg/L (starting with the known minimum inhibitory concentration on PDB) were added. In the control group, 5 mL sterile water and a spore suspension of *A. carbonarius* H9 with a concentration of 1 × 10^4^ CFU/mL were added. The inoculated tea samples were cultured in an incubator at 25 °C. Samples were taken every 5 d for a total of 5 times. The inhibition of polypeptides on *A. carbonarius* H9 was assessed visually by evaluating the fungal growth of *A. carbonarius* H9 inside and outside the tea, and the number of fungal colonies. The enumeration of fungal colonies (*A. carbonarius*) was carried out by the dilution plating method [27]. The lowest inhibitory concentration of anti-fungal peptide that was still able to inhibit the growth of *A. carbonarius* H9 for 5 d was taken as the minimum inhibitory concentration (MIC) of the anti-fungal peptides on the tea.

The ochratoxin A content on the tea samples was assessed using the method described by Mario et al. (2009) [28] with some modifications (as described below). The detection of OTA content was performed by using an Agilent 2010 high-performance liquid chromatograph (HPLC). The stationary phase used an Agilent HC-C18 column (Agilent Technologies, CA, USA, 4.6 × 250 mm; 5 μm), and the mobile phase consisted of methanol, acetonitrile and 0.15 mol/L phosphoric acid (1/1/1, *v/v/v*) at a flow rate of 0.8 mL/min. The injection volume was 20 μL, and the excitation and emission wavelength of the fluorescence detector were 300 and 460 nm, respectively.

The tea leaves inoculated with *A. carbonarius* H9 and cultured for 14 d were taken and cut into thin sections with a blade. The growth of *A. carbonarius* H9 on the tea leaves were observed by fluorescence microscope after the thin sections were stained with fungal fluorescent dye. The excitation wavelength of fungal fluorescent dye was 340~380 nm.

### 2.6. Determination of Conidial Survival Rate of A. carbonarius H9

One hundred and eighty microliters of spore suspension (1 × 10 ^3^ CFU/mL) of *A. carbonarius* H9 diluted with sterile water and 20 μL anti-fungal peptides (the inhibitory concentration obtained from the abovementioned tests in Section 2.4, sterile water used as control) were loaded into a 1.5 mL centrifuge tube, mixed evenly, and left for 16 h at 25 °C. Then 50 μL of the mixture was evenly spread on Rose Bengal medium and cultured at 25 °C for 72 h. The number of colonies formed (CFU) on the medium was counted to determine the viability of conidia. The survival rate of conidia was calculated according to Equation (1). The experiment was repeated three times at each concentration.
(1)Survival rate of conidium %=Spore number of treatmentsSpore number of control groups×100

### 2.7. Effects of Polypeptides on the Integrity of Mycelium Membrane of A. carbonarius H9

The effects of anti-fungal peptide (molecular weight from 3 to 5 kDa) on the mycelium membrane of *A. carbonarius* H9 were assessed as described by Wang et al. (2018) [26]. First, 90 μL of 1 × 10^4^ CFU/mL spore suspension of *A. carbonarius* (diluted with 5% PDB) was loaded into a 1.5 mL centrifuge tube and cultured at 25 °C. After 24 h, 10 μL of polypeptide was added to the spore suspension of *A. carbonarius* H9, reaching a final concentration of 0, 1.6 and 6.4 mg/L. The control group was treated with sterile water instead of anti-fungal peptide. After 2 h of incubation at 25 °C, 4 μmol/L SYTOX Green (SG) storage solution was added to the mixture to achieve the final SG concentration of 0.2 μmol/L. After dark treatment for 5 min, mycelia were selected and the morphological changes in mycelia were observed on the fluorescence microscope. The excitation wavelength and emission wavelength of SG were 450~490 nm and 515~565 nm, respectively. Each treatment was repeated three times.

### 2.8. Determination of Membrane Integrity of A. carbonarius H9 Spore

The effects of the polypeptide on the spore membrane of *A. carbonarius* H9 was determined according to Li et al. (2019) [29] with some modifications. First, 90 μL 1 × 10^7^ CFU/mL spore suspension of *A. carbonarius* H9 (diluted with 5% PDB) and 10 μL polypeptide were loaded into a 1.5 mL centrifuge tube, reaching final concentrations 0, 1.6 and 6.4 mg/L. The control group was treated with sterile water instead of anti-fungal peptide. After 6 h at 25 °C, 500 mg/L propidium iodide (PI) solution was added to reach a final mass concentration of PI of 50 mg/L. The samples were detected under a fluorescence microscope and photographed: the excitation wavelength and emission wavelength of PI were 535 and 615 nm, respectively, and G-2A was chosen as the optical filter. Each treatment was repeated three times. Five fields were randomly selected for each slide, and photos were taken under a bright field and fluorescent field at the same position. The number of spores observed in the bright field was defined as the total number (T), and the red-stained spores observed in the fluorescence field were regarded as the number of spores with a disrupted membrane structure (S). The integrity of the spore cell membrane was calculated according to Equation (2).
(2)Membrane integrity of spore=1−ST×100

### 2.9. Determination of Extracellular Conductivity of A. carbonarius H9

The extracellular conductivity of ochratoxigenic fungi was measured by a DDS-307A conductivity meter [30]. After 48 h culture in PDB medium, the mycelium precipitates were centrifuged at 4000× *g* for 15 min and washed three times with sterile water. The polypeptides were added to the mycelium precipitates resuspended in sterile water, reaching final concentrations 0, 1.6 and 6.4 mg/L, respectively. Sterile water instead of antimicrobial peptide solution was used as control. The extracellular conductivity was measured at 0, 1, 2, 4, 8, 16, 32 h after treatment. The experiment was repeated three times in three parallel groups at each concentration.

### 2.10. The Expression Analysis of Genes Related OTA Production in A. carbonarius H9 by RT-PCR

Total RNA was extracted using a Fungal Total RNA Isolation Kit (Sangon Biotech, Shanghai, China) according to the manufacturer’s instructions. The first strand cDNA was synthesized using the RevertAid First Strand cDNA Synthesis Kit (Thermofisher, Thermo Fisher Scientific, MA, USA). The qRT-PCR expression analysis of genes related to OTA production was carried out as described by Rachelle et al. (2017) and Ma et al. (2022) [31,32]. The primers of the targeted genes (*acpks*) and reference gene (*β-tubulin*) are listed in Table 1.

### 2.11. Statistical Analysis

Origin software (version 18.0, MicroCal, MA, USA) was used to analyze the variance. All data were expressed as the mean ± SD. All the experiments were repeated three times.

## 3. Results and Discussion

### 3.1. Anti-Fungal Peptides Produced by Brevibacillus brevis DTM05

The crude polypeptide extracts from *B. brevis* DTM05 dialyzed by a 1 kDa dialysis membrane were separated into three types of peptides with a molecular weight of MW > 5 kDa, 3 kDa ≤ MW ≤ 5 kDa and 1 kDa < MW < 3 kDa by 3 and 5 kDa ultrafiltration membranes. The ultra-filtrated polypeptides were diluted to 0.5 mg/L to fulfill the subsequent anti-fungal experiments. The results showed that the polypeptide extracts with a molecular weight of 3 kDa ≤ MW ≤ 5 kDa had the best inhibition effect on the *A. carbonarius* H9 (shown in Figure 1), and the diameter of the inhibition zone was around 25.62 ± 0.37 mm. Although the extracts of three peptides with different molecular weight showed an inhibition effect on the *A. carbonarius* H9, the peptides with a molecular weight of 3 kDa ≤ MW ≤ 5 kDa had the best anti-fungal effect, indicating that the molecular weight of peptides with an anti-fungal effect was mainly between 3 and 5 kDa.

The FT-IR spectra of the crude polypeptides with molecular weight between 3 and 5 kDa were dominated by the peaks at different wavelengths of 3908.35, 3406.39, 2918.37, 2855.62, 2364.81, 1657.87, 1542.62, 1459.88, 1051.33, 783.62, 699.28, 501.66 and 419.69 cm^−1^ (shown in Figure 2). According to the work of Long et al. [33] and Liu et al. [34], the absorption of β-sheet and α-helix were in the frequency area of 1610–1640 and 1650–1660, respectively; thus, bands around 1657.87 were expected for the crude polypeptides with α-helical and β–sheet structures, which is conducive to maintaining the secondary structure stability of polypeptides [35]. Absorbance at 3406.39 cm^−1^ wavenumbers was due to the stretching vibrations of the O–H group, indicating the presence of hydrogen bonds (intramolecular and intermolecular) in the polypeptide molecules [34]. Absorbance at 2918.37, 2855.62 and 2364.81 wavenumbers may be due to –CH_2_ asymmetric and symmetric stretching vibrations of lipid in crude polypeptides [34]. The rest of the peaks at 1522.62, 1459.88, 1051.33, 783.62, 699.28, 501.66 and 419.69 cm^−1^ were caused by CH_2_ scissoring, CH_3_ bending, and C–O stretching vibration, respectively [36,37,38,39]. Therefore, it can be speculated that the crude peptide was a mixture of mainly polypeptides and small amounts of lipids and other carbohydrates.

### 3.2. Inhibition Effects of the Polypeptides on the Growth of A. carbonarius H9

Different concentrations of the polypeptides were added into the spore suspension of *A. carbonarius* H9, and the differences between the initial and terminal (cultured after 48 h) absorbance at 600 nm (ΔOD600) of the suspension were analyzed to evaluate spore germination rate and the growth rate of mycelium. In general, the ΔOD600 value of *A. carbonarius* H9 gradually decreased with the increase in the polypeptide concentration. The ΔOD600 value of the suspension of *A. carbonarius* H9 was close to zero when the polypeptides concentration was 1.6 mg/L, indicating that the growth of *A. carbonarius* H9 spores was completely inhibited. Therefore, it can be inferred that the MIC of anti-fungal peptides on *A. carbonarius* H9 was 1.6 mg/L (as shown in Figure 3a). The germination rate of conidia of *A. carbonarius* H9 treated with polypeptides was significantly reduced, that is, the polypeptides significantly affected the survival rate of spores as shown in Figure 3b and Figure 4. However, the germination rate of spores treated by polypeptides with concentrations of 1.6 mg/L was still close to 20%. The germination rate did decrease to about 4% but only after treatment with a high concentration (25.6 mg/L) of the polypeptides. The polypeptides can inhibit the growth of *A. carbonarius* H9 in vitro (MIC was 1.6 mg/L) and polypeptides with a concentration of MIC can also significantly reduce the germination rate of *A. carbonarius* H9 spore to 20.62%, but they cannot completely kill all spores of *A. carbonarius* H9. These results were consistent with [40], which reported the anti-fungal effect of the fermentation supernatant of *B. cereus* on *A. niger*, and with [15], which involved the inhibitory effect of the polypeptide PAF26 on the growth and spore germination of *Monilinia fructicola*.

### 3.3. Inhibitory Effects of Polypeptides on the Growth of A. carbonarius H9 on Tea Matrix

The sterilized tea samples with different concentrations of polypeptides were used as culture substrate and the *A. carbonarius* H9 isolated from the post-fermented tea was used as the indicator fungus to assess the inhibitory effects of crude polypeptides on the ochratoxigenic fungi. The same concentration of *A. carbonarius* H9 spore was inoculated into the center of the tea matrix and the diameter of the fungal plaque and OTA content in the contaminated tea samples were analyzed; the results are shown in Figure 5. The lowest concentration of peptides that significantly inhibited the growth of *A. carbonarius* H9 on the tea matrix was 3.2 mg/L. When the concentrations of crude polypeptides in the tea matrix was 6.4 mg/L, the fungal plaque diameter on the tea matrix was significantly less than that of the control group (shown in Figure 5a,b). Although the fungal plaque diameter on the tea matrix increased with the extension of culture time, the fungal plaque diameter decreased with the increase in the polypeptide concentration in the tea samples, which indicated that crude polypeptide (with concentration of 3.2, 6.4 mg/L in the tea matrix) had obvious inhibitory effects on the growth of *A. carbonarius* H9.

The fluorescence micrograph of the tea leaves inoculated with *A. carbonarius* H9 is shown in Figure 5b. After 14 d of incubation, as for the control group, a large amount of mycelium of *A. carbonarius* H9 grew inside and outside the leaf tissue. Although a large number of hyphae of *A. carbonarius* H9 grew on the surface of tea leaves treated with 3.2 mg/L polypeptide, only a small amount of mycelium penetrated into the internal tissue of the tea leaves. For tea leaves treated with 6.4 mg/L polypeptide, fungal mycelium could hardly be seen, either on the leaf surface or inside leaf tissues. The results of the fluorescence microscopy showed that polypeptide (with a concentration of 3.2, 6.4 mg/L in the tea matrix) had significant inhibitory effects on the growth of *A. carbonarius* H9 on the tea matrix, and the inhibitory effect of 6.4 mg/L polypeptide was greater than that of 3.2 mg/L.

The OTA contents in tea samples spiked with a concentration of 3.2 and 6.4 mg/L were lower than those of the control, which means that polypeptides had an obvious inhibitory effect on the OTA production of *A. carbonarius* H9 with an inhibition rate ranging from 57.09 ± 0.15% to 92.66 ± 0.78% after 14 days of incubation, for an OTA concentration of 15.46 μg/kg and 1.37 μg/kg on the tea matrix, respectively (Figure 5d). *B. brevis* has been reported to inhibit the growth and OTA production of *A. carbonarius* H9 on tea substrates [10]. This study further revealed that the polypeptides produced by *B. brevis* had significant inhibitory effects on the growth and OTA production of *A. carbonarius* H9. At the same time, it should be noted that a higher concentration of polypeptides may be required to exert the same inhibitory effect on the growth of *A. carbonarius* H9 on the tea substrate compared with the pure culture, which implies a decrease in the effective antimicrobial concentration of the peptides on the tea substrate. Because our previous studies had shown that temperature (20~60 °C) and acidic environments (pH 5 to 7) did not significantly affect the anti-fungal activity of the polypeptides (unpublished data), the results are similar to the effects of temperature and pH on the antibacterial activity of the fermentation supernatant of *Bacillus cereus* found by Yi et al. (2022) [40]. Therefore, it is speculated that the strong absorbability of tea [41] may be an important reason for reducing the effective antibacterial concentration of polypeptides on tea substrate, or there are other factors affecting the antibacterial activity of polypeptides on tea substrates that need to be further studied.

### 3.4. Effect of Polypeptides on Spore Cell Membrane Integrity of A. carbonarius H9

The PI fluorescent stain can pass through the damaged cell membrane into the cells and combine with the nucleic acid material to emit red fluorescence when the cell membrane of *A. carbonarius* H9 spores is damaged [42]. No cell-emitted red fluorescence was observed in the control spore cells without polypeptides treatment, but a certain number of cells were observed to emit red fluorescence in spore cells treated by polypeptides with concentrations of 1.6, 6.4 mg/L, as shown in Figure 6. The integrity of the spore cell membrane of *A. carbonarius* H9 without polypeptide treatment was maintained at 100% (control group), while the membrane integrity of spore cells treated with 1.6 and 6.4 mg/L was significantly reduced. The higher the concentration of polypeptides, the more severe the disruption to the cell membrane integrity of *A. carbonarius* H9. When the polypeptide concentration of treated *A. carbonarius* H9 spores was 6.4 mg/L, the integrity of the cell membrane was only 12.29% (as shown in Figure 7). It is generally believed that the inhibition of polypeptide on the fungal spore survival rate is mainly manifested through the destruction of spore cell membranes (Qian et al., 2015; Cai et al., 2020). The polypeptides with antifungal effects such as Bacillomycin D (a polypeptide derived from *B. subtilis*) [42], PAF26 [15], etc., were found to have significant inhibitory and destructive effects on fungal spore cell membranes. It is possible that antifungal peptides may destroy the spore cell membrane, leading to the leakage of nucleic acids and proteins, and to the production of reactive oxygen species that cause apoptosis of spore cells, which eventually inhibits germination and the survival rate of fungal spore [26,29].

### 3.5. Effect of Anti-Fungal Peptides on Mycelia Cell Membrane Integrity of A. carbonarius H9

When the permeability of mycelia cell membrane is destroyed, SG fluorescent stain penetrates into the cell and is bound with nucleic acid materials to present bright green fluorescence [15]. No green fluorescence was observed when the mycelia of *A. carbonarius* H9 were not treated with the polypeptides (Figure 8A_1_,A_2_). The mycelium of *A. carbonarius* H9 treated by polypeptides at 1.6 mg/L showed a weak green fluorescence (Figure 8B_1_,B_2_). It was interesting that when the mycelia of *A. carbonarius* H9 was exposed to polypeptides at 6.4 mg/L, almost more than 95% of the mycelium were found to emit continuous, bright green fluorescence. This suggests that the permeability of mycelium cell membrane increased with the increase in polypeptide concentration, which led to the increase in SG fluorescence in mycelia cells. Cell membrane is an important barrier to protect cells and maintain the balance of osmotic pressure inside and outside the cell. Once the permeability of the cell membrane is destroyed by polypeptides, cell death will occur [29]. The fluorescence of mycelium was significantly enhanced after polypeptide treatment, which indicated that the anti-fungal effect of polypeptides produced by *B. brevis* DTM05 was exerted by changing the cell permeability. This is very similar to the antifungal mode of most antimicrobial peptides reported in the literature, such as PAF56 [26].

### 3.6. Effect of Anti-Fungal Peptides on Extracellular Conductivity of A. carbonarius H9

The mycelium extracellular conductivity of *A. carbonarius* H9 treated by polypeptides with different concentrations was analyzed, as shown in Figure 9. The extracellular conductivity of *A. carbonarius* H9 mycelia in both control and treatment groups increased gradually with the prolongation of the treatment time. The mycelium extracellular conductivity of *A. carbonarius* H9 treated by polypeptides was higher than that of the control group and the conductivity of the highest concentration (6.4 mg/L) treatment was significantly higher than that of the lowest concentration (1.6 mg/L).

Electrical conductivity is an important index to measure the change in cell permeability. When the cell permeability increases, intracellular substances such as proteins, sugars, etc., will leak out, eventually leading to an increase in the electrical conductivity indicators [43]. The extracellular conductivity of *A. carbonarius* H9 mycelium treated by polypeptides from *B. brevis* DTM05 increased significantly and promoted the leakage of organic substances inside the mycelium membrane. This result indicated that the mycelium membrane was damaged by polypeptides from *B. brevis* DTM05 to a certain extent, which is similar to the effects of other polypeptides such as Bacillomycin D [43], PAF56 [26] and PAF26 [15].

### 3.7. Effect of Anti-Fungal Peptides on the Expression of Genes Related to OTA Production in A. carbonarius H9

OTA biosynthesis is mainly regulated by the polyketide synthase gene (*acpks* and *acOTApks*), non-ribosomal peptide synthase gene family (*acOTAnrps*) and other regulators (*laeA* and *veA*) [31,43,44,45]. In order to further clarify the inhibition effects of polypeptides from *B. brevis* DTM05 on *A. carbonarius* H9, the effects of polypeptides on the expression of OTA polyketide synthase gene acpks in *A. carbonarius* H9 were investigated. The results showed that the polypeptides significantly attenuated the polyketide synthase gene *acpks* expression levels (Figure 10). The higher the concentration of polypeptides, the more the expression levels of the polyketide synthase gene *acpks* were down-regulated. When the concentration of polypeptides was 6.4 mg/L, the relative expression level of *acpks* was down-regulated by 81.6% (*p* < 0.01). The inhibition mechanism of polypeptides from *B. brevis* DTM05 in the OTA synthesis of *A. carbonarius* H9 may be similar to that of Bacillomycin D produced by *B. subtilis* [43] as both of them reduce OTA synthetase activity by down-regulating the expression level of the polyketide synthase gene *acpks*, thus eventually reducing OTA production.

## 4. Conclusions

The molecular weight of antimicrobial polypeptides produced by *B. brevis* DTM05 mainly ranged between 3 and 5 kDa. The FT-IR spectra of polypeptides (with a molecular weight between 3 and 5 kDa) showed that it was a mixture consisting mainly of polypeptides, and small amounts of lipids and other carbohydrates. The polypeptide produced by *B. brevis* DTM05 inhibited the growth of *A. carbonarius* by increasing the permeability of the mycelium membrane and destroying the *A. carbonarius* H9 spore membrane, resulting in the leakage of intracellular substances and the death of fungal cells. The study also confirmed that polypeptide (with a concentration ≥6.4 mg/L) significantly reduced the OTA production of *A. carbonarius* H9 on the tea matrix by down-regulating the expression level of the polyketide synthase gene *acpks* related to OTA production. In conclusion, the polypeptides produced by *B. brevis* DTM05 (isolated from post-fermented tea samples) showed significant antifungal activity against *A. carbonarius* H9 (one of the ochratoxigenic fungi often detected in the pile-fermentation process of post-fermented tea), and it is expected to be a potential and effective way to control the contamination of ochratoxigenic fungi and OTA production during the pile-fermentation process of post-fermented tea. For practical applications, the stability and antimicrobial activity of the polypeptides in the tea matrix during the pile-fermentation process under complex microecological conditions need to be further studied.

## Figures and Tables

**Figure 1 foods-11-03243-f001:**
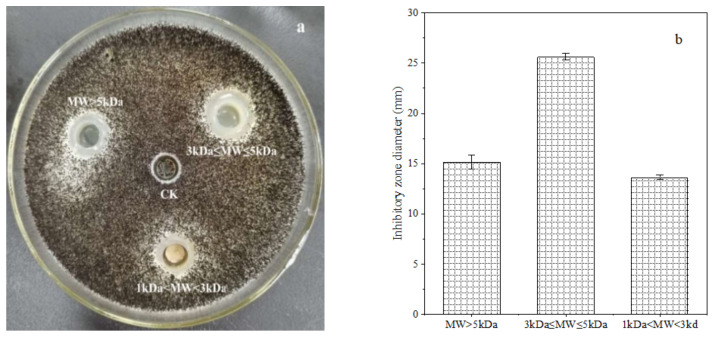
Inhibition effect of polypeptides (at a concentration of 0.5 mg/L) with different molecular weight (0.5 mg/L) produced by B. brevis DTM05 on *A. carbonarius* H9 (**a**). The inhibition zone diameter of polypeptides with different molecular weight on *A. carbonarius* H9 (**b**).

**Figure 2 foods-11-03243-f002:**
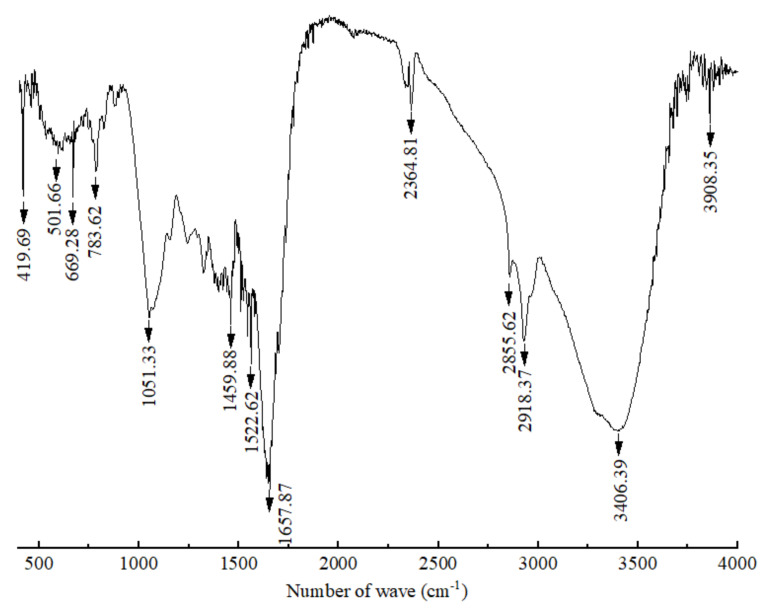
Infrared spectrum of polypeptides with a molecular weight between 3 and 5 kDa produced by *B. brevis* DTM05.

**Figure 3 foods-11-03243-f003:**
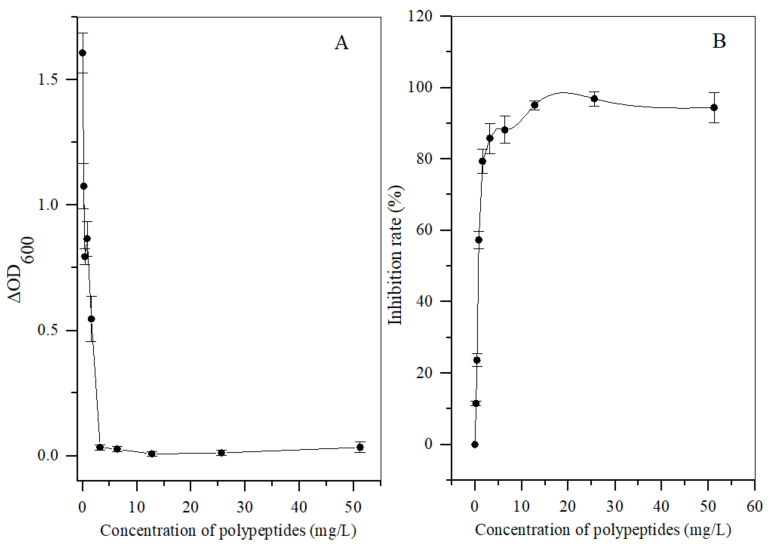
Antifungal activity in vitro of polypeptides with a molecular weight from 3 kDa to 5 kDa produced by *B. brevis* DTM05 on *A. carbonarius* H9. (**A**) effects of the polypeptides with different concentrations on the spore growth of *A. carbonarius* H9; (**B**) effects of the polypeptides with different concentrations on the spore survival rate of *A. carbonarius* H9).

**Figure 4 foods-11-03243-f004:**
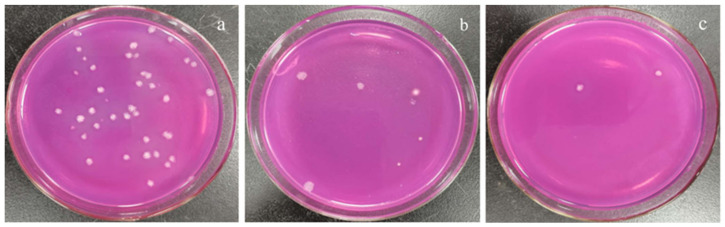
Effect of polypeptides treatment on spore germination rate of *A. carbonarius* H9 spores. (**a**) 1.6 mg/L polypeptides; (**b**) 3.2 mg/L polypeptides; (**c**) 25.6 mg/L polypeptides.

**Figure 5 foods-11-03243-f005:**
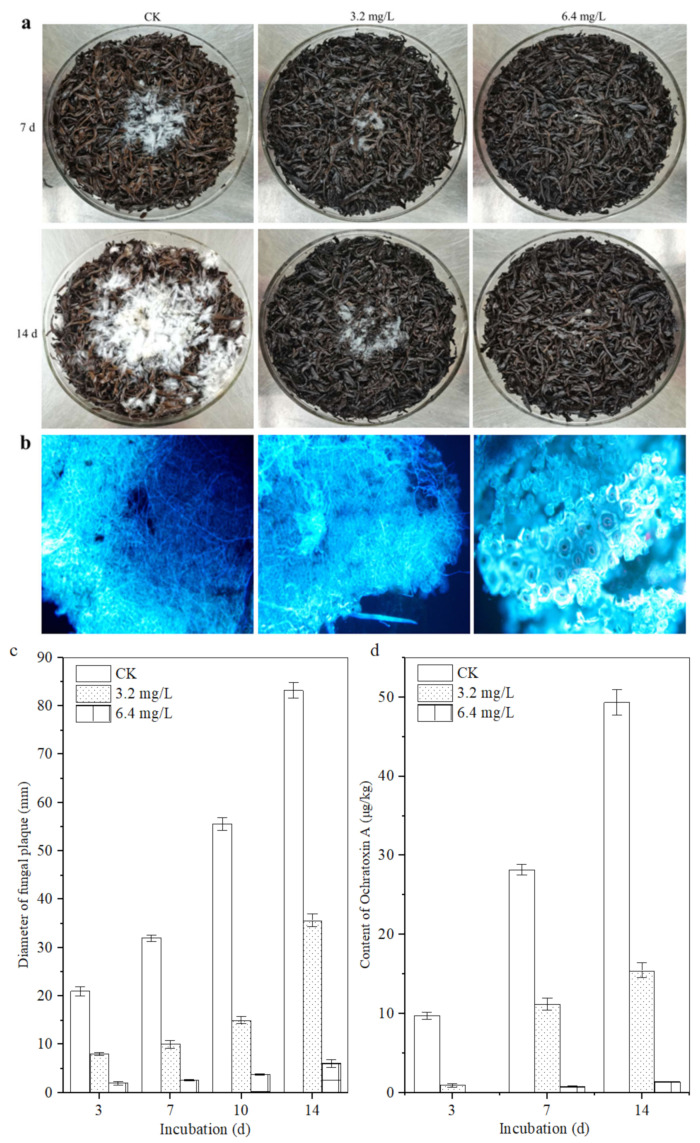
Inhibitory effect of polypeptides on the growth of *A. carbonarius* H9 on the sterilized tea. (**a**) Visible growth of *A. carbonarius* H9 on the tea substrate; (**b**) fluorescence microsections of tea samples treated by *A. carbonarius* H9 for 14 days; (**c**) the fungal plaque diameter on the tea matrix; (**d**) the inhibitory effects of crude polypeptides on OTA content in the contaminated tea samples.

**Figure 6 foods-11-03243-f006:**
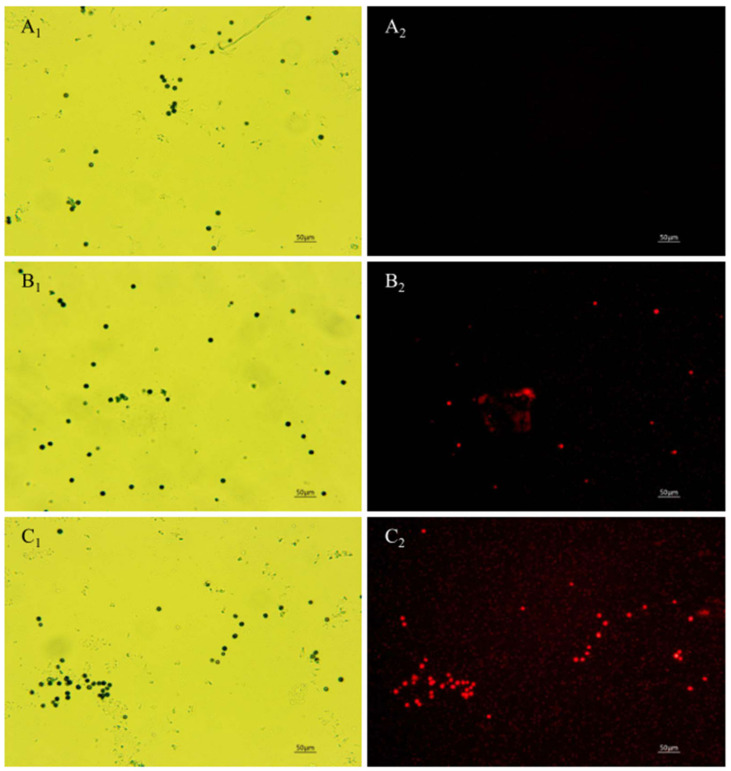
Effect of anti-fungal peptides treatment on membrane permeability of *A. carbonarius* H9 spores. (**A_1_**,**A_2_**) control group under bright field and fluorescent field; (**B_1_**,**C_1_**) spore membrane treated by 1.6 and 6.4 mg/L polypeptides under bright fields; (**B_2_**,**C_2_**) spore membrane treated by 1.6 and 6.4 mg/L polypeptides under fluorescent fields.

**Figure 7 foods-11-03243-f007:**
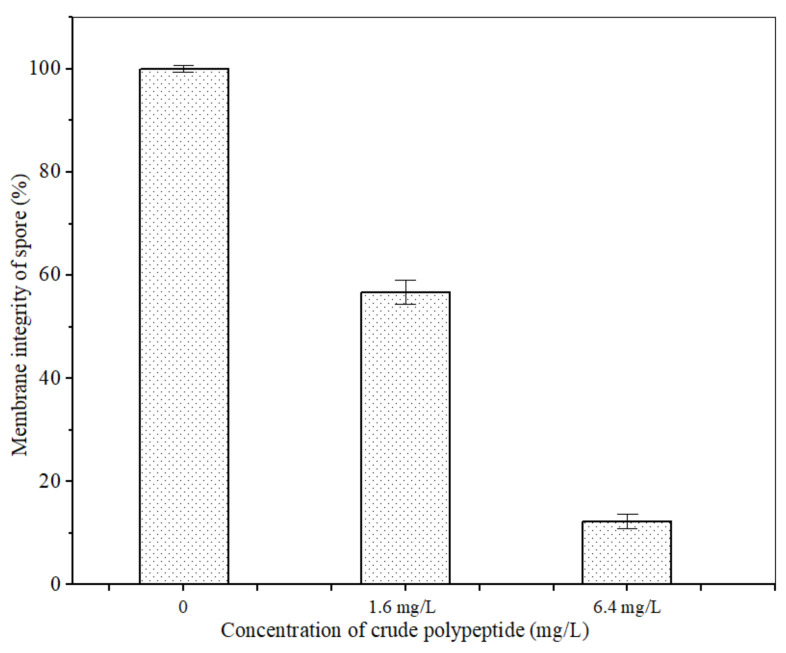
Effect of anti-fungal peptides on membrane integrity of *A. carbonarius* H9 spores.

**Figure 8 foods-11-03243-f008:**
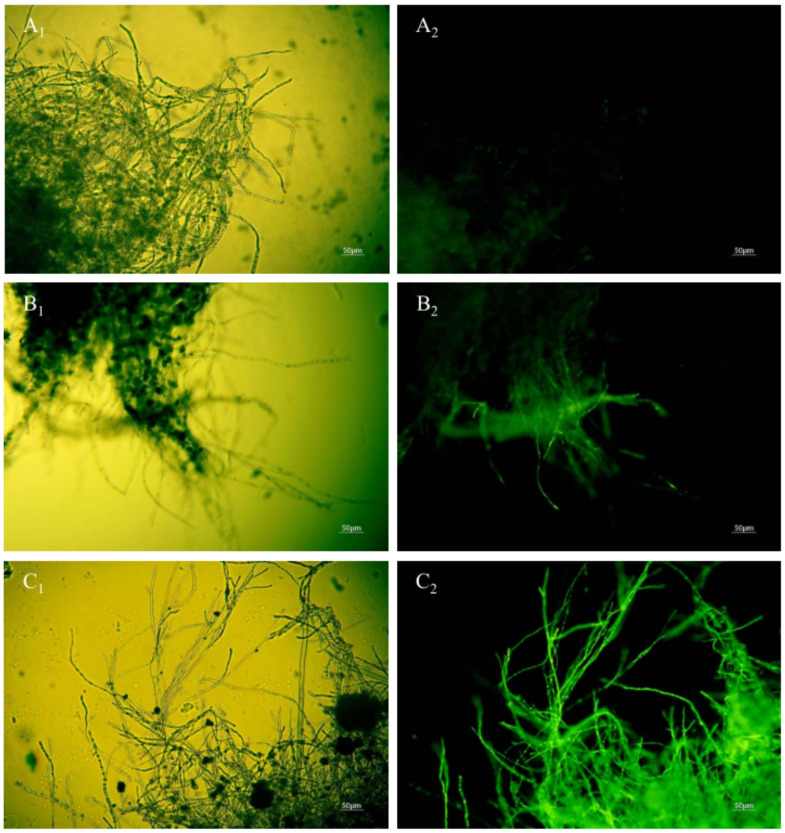
Effect of anti-fungal peptides treatment on membrane permeability of A. carbonarius H9 mycelia (**A_1_**,**A_2_**), control group under bright field and fluorescent field; (**B_1_**,**C_1_**), mycelia membrane treated by 1.6 and 6.4 mg/L polypeptides under bright field; (**B_2_**,**C_2_**), mycelia membrane treated by 1.6 and 6.4 mg/L polypeptides under fluorescent field).

**Figure 9 foods-11-03243-f009:**
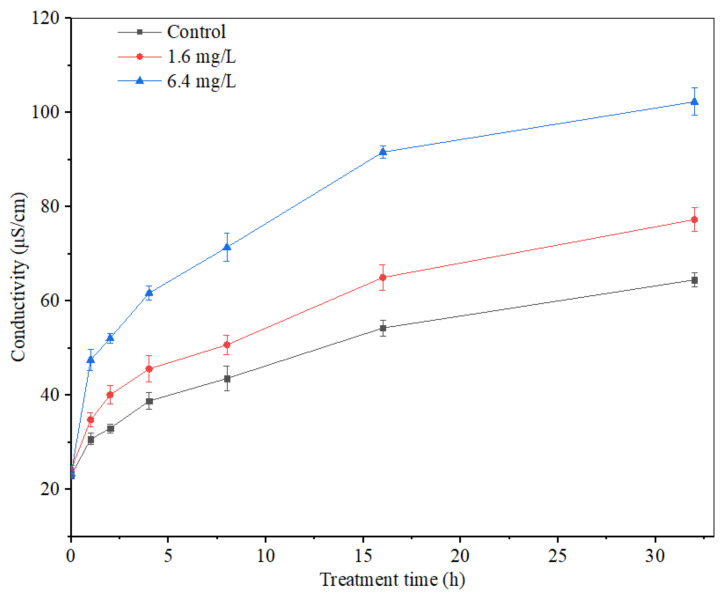
Effect of anti-fungal peptides on mycelium extracellular conductivity of *A. carbonarius*.

**Figure 10 foods-11-03243-f010:**
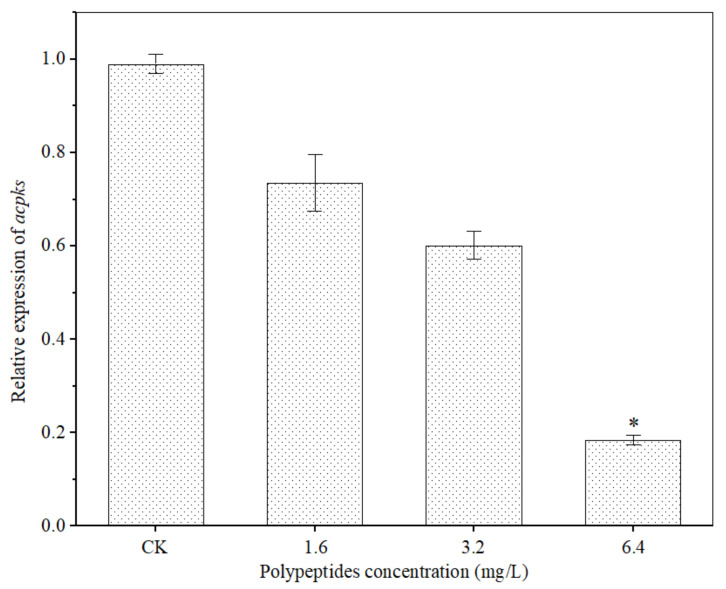
Effects of polypeptides on expression of the OTA polyketide synthase gene *acpks* in *A. carbonarius*. *, significant difference (*p* < 0.01).

**Table 1 foods-11-03243-t001:** The primers used for quantitative RT-PCR.

Primer Name	Primer Sequence (5′–3′)	Reference
*acpks-F*	GAGTCTGACCATCGACACGG	[31]
*acpks-R*	GGCGACTGTGACACATCCAT	

## Data Availability

The data resulting from this study are available within the article.

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
