# Peer review of "Inhibitory Effect of Polypeptides Produced by Brevibacillus brevis on Ochratoxigenic Fungi in the Process of Pile-Fermentation of Post-Fermented Tea"

_foods, 2022, doi:10.3390/foods11203243_

Round 1

Reviewer 1 Report

Abstract

Provide full form of OTA

Line 34: rephrase: Fluorescence staining results showed that the polypeptides could enhance -----.

Line 39-42: Rewrite and give a strong conclusion of the present findings.

Introduction:

Line 59: delete ‘and’

Line 70: correct: Aspergillus and Penicillium

Line 76: expand CYA

Line 98: correct: B. coagulans and B. laterosporus etc

Materials and methods

Section 2.3: Provide uniform pattern of units kD or kDa?

Section 2.5: Mention the different concentrations of antimicrobial peptides used.

How the molecular weight of polypeptides were determined?

Whether performed SDS-PAGE analysis?

Conclusion

Rewrite this part. Provide overall conclusion about how this study would benefit the society. How the produced polypeptides are economical for use. How it is superior to commercial antimicrobial drugs? Add this information.

Reference

Some of the cited references are outdated. Replace with recent references.

Author Response

Answers for Reviewer (1)- Queries
Abstract 

Question 1:  Provide full form of OTA

Answer: The full form (ochratoxin A) of OTA has been provided in line 32.

Question 2: Line 34: rephrase: Fluorescence staining results showed that the polypeptides could enhance -----.

Answer:  The mentioned result has been rephrased in line 34-39: “The enhancement of fluorescence staining signal in the mycelium and conidiospore of A. carbonarius H9 showed that the polypeptides with concentration more than 1.6 mg/L could increase the permeabilities of mycelium membrane and conidial membrane of A. carbonarius H9. The significant increase of extracellular conductivity of A. carbonarius H9 mycelia meant the outward leakage of intracellular living substance and also further indicated the increase of cell membrane permeability.”

Question 3: Line 39-42: Rewrite and give a strong conclusion of the present findings.”

Answer: This sentence has been rewritten as suggested by reviewer in lines 41-45: “In conclusion, the polypeptides produced by Brevibacillus brevis can destroy the structural integrity of A. carbonarius cell membrane, make the outward leakage of intracellular living substance, accelerate the death of A. carbonarius, and then effectively control the contamination of ochratoxingenic fungi and OTA production during the pile-fermentation process of the post-fermented tea.”

Introduction:

Question 4: Line 59: delete ‘and’

Answer: “and” has been deleted in line 62 of revised manuscript.

Question 5: Line 70: correct: Aspergillus and Penicillium

Answer: “Aspergillus and Penicillium” have been revised in line 62 of revised manuscript.

Question 6: Line 76: expand CYA

Answer:“Czapek yeast exatract agar” has been added in line 81 of revised manuscript.

Question 7: Line 98: correct: B. coagulans and B. laterosporus etc

Answer:B. coagulans and B. laterosporus etc” has been revised in line 104 of revised manuscript.

Materials and methods

Question 8: Section 2.3: Provide uniform pattern of units kD or kDa?

Answer: The full name for molecular weight unit is kDa and often is abbreviated as kD. kD has been completely revised to kDa in this paper as suggested by reviewer.

Question 9: Section 2.5: Mention the different concentrations of antimicrobial peptides used.

Answer: The concentrations of antimicrobial peptides (1.6, 3.2, 6.4, 12.8, 25.6 mg/L) has added in line 198-199 of revised manuscript. The concentrations were set according to the known minimum inhibitory concentration on PDB from Section 2.4.

Question 10: How the molecular weight of polypeptides were determined?

Answer: Polyeptides were ultrafiltered by Ultrafiltration Centrifugal Tube of 3 kDa and 5 kDa (Millipore, Germany) and separated into peptides with molecular weights less than 3kDa, 3kDa-5kDa and more than 5kDa, respectively.

Question 11: Whether performed SDS-PAGE analysis?

Answer: SDS-PAGE analysis was performed, but the picture of SDS-PAGE was not clear enough, so it was not mentioned in this article.

Conclusion

Question 12: Rewrite this part. Provide overall conclusion about how this study would benefit the society. How the produced polypeptides are economical for use. How it is superior to commercial antimicrobial drugs? Add this information.

Answer:

The conclusion has been rewritten as suggested by reviewer: “The molecular weight of antimicrobial polypeptides produced by Br. Brevis DTM05 was mainly distributed between 3 kDa and 5 kDa. The FT-IR spectra of polypeptides (molecular weight between 3 kDa and 5 kDa) showed that it was a mixture consisting of mainly polypeptides, small amounts of lipids and other carbohydrates. The polypeptide produced by Br. Brevis DTM05 inhibited the growth of A. carbonarius by increasing the permeability of mycelium membrane and destroying A. carbonarius H9 spore membrane, resulting in the leakage of intracellular living substance and the death of fungal cell. The study also confirmed that polypeptide (with the concentration 6.4 mg/L) significantly reduced OTA production of A. carbonarius H9 on tea matrix by down-regulating the expression level of the polyketide synthase gene acpks related to OTA production. In conclusion, the polypeptides produced by Br. brevis DTM05 (isolated from post-fermented tea samples) showed significantly antifungal activity against A. carbonarius H9 (one of ochratoxingenic fungi often detected in the pile-fermentation process of post-fermented tea), and it was expected to be a potential and effective way to control the contamination of ochratoxingenic fungi and OTA production during the pile-fermentation process of the post-fermented tea. For the practical applications, the stability and antimicrobial activity of the polypeptides in tea matrix during the pile-fermentation process under complex microecological conditions need to be further studied.”

Reference

Question 13: Some of the cited references are outdated. Replace with recent references.

Answer: Some of the references have been replaced with recent references.

Reviewer 2 Report

Title of Article: Inhibitory effect of polypeptides produced by Brevibacillus 2 brevis on ochratoxingenic fungi in the process of pile-fermenta- 3 tion of post-fermented tea

Manuscript Number: Foods-1882739

The paper has lots of work and experimental design is also good. Results are also impressive. But writing part is poor. English language, technical writing should be improved. Some points are pointed out and they should be applied in whole manuscript.

Line 156 and 160: Br. Brevis

Line 157 and 159: use same pattern 4℃, 37 °C

Line 208-209: 330 and 460 nm, respectively.

In figure 5 you used 3.2 mg/L concentration, after it in Figure 7 and 8 you used 1.6 mg/L. Why?

    General comment: Introduction part is very long, this much is not required.

Capital words in between lines

Grammatically paper is good but typographical mistakes are high.

Scientific names in Italics

Author Response

Answers for Reviewer (2)- Queries

Question 1: The paper has lots of work and experimental design is also good. Results are also impressive. But writing part is poor. English language, technical writing should be improved. Some points are pointed out and they should be applied in whole manuscript.

Answer: We have consulted senior editor to improve the English language level of our manuscript, and made many changes, see the revised text.

Question 2: Line 156 and 160: Br. Brevis

Answer: Br. Brevis has been modified to Br. Brevis in the whole revised manuscript.

Question 3: Line 157 and 159: use same pattern 4℃, 37 °C

Answer: The pattern has been changed to the same.

Question 4: Line 208-209: 330 and 460 nm, respectively.

Answer: “nmhas been deleted.

Question 5: In figure 5 you used 3.2 mg/L concentration, after it in Figure 7 and 8 you used 1.6 mg/L. Why?

 Answer: Because the lowest concentration of polypeptides that significantly inhibited the growth of A. carbonarius H9 was 3.2 mg/L on tea matrix, and polypeptide (with the concentration of 1.6 mg/L) had a good antibacterial effect in the PDB medium.

Question 6: General comment: Introduction part is very long, this much is not required.

Answer: Introduction part has been cut as suggested by reviewer.

Question 7: Grammatically paper is good but typographical mistakes are high.

Answer: Typographical mistakes have been carefully corrected.

Question 8: Scientific names in Italics

Answer: Species Latin names have been changed to italics in the whole revised manuscript.

Reviewer 3 Report

The manuscript entitled “Inhibitory effect of polypeptides produced by Brevibacillus brevis on ochratoxingenic fungi in the process of pile-fermentation of post-fermented tea” is well written but it requires revision on the points mentioned below.

1)     Name of microorganisms and fungus should be italic, but I have found several such incidents in the manuscript that those names are not italic (examples lines 277/283/285/332 etc.).

2)     Please show the SDS-PAGE picture of anti-fungal polypeptides of 3 kDa to 5 kDa. Did you mention the PDB id of those proteins? Or accession number of genes?

3)     Usually, the gene name is not shown as in Table 1. Please follow the literature.

4)     Line 296, it is not molecular weight. It’s a mass.

5)     Line 314-315: I did not find your explanation on α-helix and β-sheet with wave numbers of FT-IR in cited paper [32].

6)     You have shown the effect of the polypeptide on the spore germination of fungus. What is the mechanism of such an effect? Show your hypothesis in graphical representation.

7)     Please lower the similarity index in the manuscript significantly.

Author Response

Answers for Reviewer (3)- Queries

Question 1: Name of microorganisms and fungus should be italic, but I have found several such incidents in the manuscript that those names are not italic (examples lines 277/283/285/332 etc.).

Answer: Species Latin names have been changed to italics as suggested by reviewer in the whole revised manuscript.

Question 2:  Please show the SDS-PAGE picture of anti-fungal polypeptides of 3 kDa to 5 kDa. Did you mention the PDB id of those proteins? Or accession number of genes?

Answer: Polyeptides were ultrafiltered by Ultrafiltration Centrifugal Tube of 3 kDa and 5 kDa (Millipore, Germany) and separated into peptides with molecular weights less than 3kDa, 3kDa-5kDa and more than 5kDa, respectively.

Although SDS-PAGE analysis was performed, the picture of SDS-PAGE was not clear enough, so it was not mentioned in this article.

Figure 1. SDS-PAGE analysis polypeptide extracts of Br. Brevis.

PDB mentioned in paper is only a medium of potato dextrose broth (PDB).

Question 3:  Usually, the gene name is not shown as in Table 1. Please follow the literature.

Answer: The description of gene primer was modified according to the reference [30] in line 282-283 of revised manuscript.

Question 4:   Line 296, it is not molecular weight. It’s a mass.

Answer: The description is a little unclear, and it has been revised into “Inhibition effect of polypeptides (at concentration of 0.5 mg/L) with different molecular weight”

 Question 5:   Line 314-315: I did not find your explanation on α-helix and β-sheet with wave numbers of FT-IR in cited paper [32].

Answer: The description of this part was rewritten as follows: “According to the assignments of Long [32] and Liu [33], the absorption of β-sheet and α-helix were in frequency area of 1610-1640 and 1650-1660, respectively, bands around 1657.87 were expected for the crude polypeptides with α-helical and β-sheet structures, which was conducive to maintain the secondary structure stability of polypeptides [34].”

 Question 6: mechanism of such an effect? Show your hypothesis in graphical representation.

Answer: The mechanism of action of polypeptide against A. carbonarius is by destroying the integrity of the cell membrane, enhancing the permeability of the membrane, leading to the leakage of internal substances, and finally leading to cell death. The mechanism for the reduction of OTA production is that polypeptides were involved in the down-regulation of the expression level of the polyketide synthase gene acpks related to OTA synthesis.

 Question 7:  Please lower the similarity index in the manuscript significantly.

Answer: The similarities with other articles have been modified in the revised manuscript.

Round 2

Reviewer 3 Report

Dear authors,

I had asked you to provide the PDB (Protein database) id of small peptides (if available) Or the accession number of those genes. Probably you did not understand my comments clearly. During proofreading, you can correct those things.

Author Response

Thank you very much for your encouragements and constructive comments on the revision of our manuscript.

Question 1:  I had asked you to provide the PDB (Protein database) id of small peptides (if available) Or the accession number of those genes. Probably you did not understand my comments clearly. During proofreading, you can correct those things.

Answer: This study mainly reported the anti-fungal effect of polypeptide (molecular weight from 3kDa to 5kDa) produced by B. brevis on the ochratoxigenic fungus A. carbonarius. The identification of antifungal polypeptides is about to be completed, and PDB (Protein database) id will be reported in the subsequent paper. Please continue to pay attention to our research, thank you!
